# Distinctive Deposition Patterns of Sporadic Transthyretin-Derived Amyloidosis in the Atria: A Forensic Autopsy-Based Study

**DOI:** 10.3390/ijms25158176

**Published:** 2024-07-26

**Authors:** Shojiro Ichimata, Yukiko Hata, Koji Yoshida, Keiichi Hirono, Naoki Nishida

**Affiliations:** 1Department of Legal Medicine, Faculty of Medicine, University of Toyama, Toyama 930-0194, Japannishida@med.u-toyama.ac.jp (N.N.); 2Department of Pediatrics, Faculty of Medicine, University of Toyama, Toyama 930-0194, Japan; khirono@med.u-toyama.ac.jp

**Keywords:** atrium, atrial natriuretic factor, cardiac amyloidosis, giant cell, transthyretin

## Abstract

Left-to-right differences in the histopathologic patterns of transthyretin-derived amyloid (ATTR) deposition in the atria of older adults have not yet been investigated. Hence, this study evaluated heart specimens from 325 serial autopsy subjects. The amount of ATTR deposits in the seven cardiac regions, including both sides of atria and atrial appendages, was evaluated semiquantitatively. Using digital pathology, we quantitatively evaluated the immunohistochemical deposition burden of ATTR in the myocardium. We identified 20 sporadic ATTR cardiac amyloidosis cases (nine males). All patients had ATTR deposition in the left atrial regions of the myocardium. In the semiquantitative analysis, 14 of the 20 cases showed more severe ATTR deposition on the left atrial regions than on the right side, with statistically significant differences in the pathology grading (*p* < 0.01 for both the atrium and atrial appendage). Quantitative analysis further supported the difference. Moreover, six had ATTR deposition in the epineurium and/or neural fibers of the atria. Cluster analysis revealed that ATTR deposition in the myocardium was significantly more severe in males than in females. The heterogeneous distribution of amyloid deposits between atria revealed in this study may impair the orderly transmission of the cardiac conduction system and induce arrhythmias, which may be further aggravated by additional neuropathy in the advanced phase. This impairment could be more severe among males. These findings emphasize that atrial evaluation is important for individuals with sporadic ATTR cardiac amyloidosis, particularly for early detection.

## 1. Introduction

Amyloidosis is classified according to the precursors of amyloid fibril proteins deposited in tissues [1]. Although various symptoms can arise from amyloid deposition, cardiac amyloidosis (CA) is the primary cause of death among patients with systemic amyloidosis [2]. CA primarily results from the following two forms of precursor proteins: transthyretin-derived amyloid (ATTR) and immunoglobulin light chain-derived amyloid [3]. Recently, the importance of diagnosing ATTR-CA has been exponentially recognized because of advances in diagnostic and therapeutic methods [4,5]. Early diagnosis is necessary, given that the therapeutic outcome of drugs that stabilize transthyretin is more effective when treatment is administered early [6,7]. Therefore, the histopathological progression pattern in patients with ATTR-CA should be comprehensively understood to establish more effective diagnostic approaches.

In recent years, the evaluation of ATTR deposition in the atria has been given increased attention [8,9,10,11]. We previously found that ATTR deposition patterns within the atrial septum (AS) differed on each side [12]. Consequently, the distribution and severity of ATTR deposition may differ between the left and right atrial regions, including the atrium and atrial appendage. Although atrial natriuretic factor-derived amyloid (AANF) is commonly found in the atria of nearly all old-aged patients with a left-predominant deposition [13,14,15], the deposition pattern of ATTR in the left and right atrial regions remain to be investigated. Furthermore, several studies have reported the potential colocalization of ATTR and AANF [8,11,12,16]. However, the relationship between ATTR and AANF is yet to be elucidated.

We previously reported that patients with sporadic ATTR-CA (sATTR-CA) could be at risk for sudden cardiac death (SCD), even from an early phase [12]. Thus, apart from the severity of ATTR deposition in the myocardium, additional risk factors for SCD might be involved. ATTR deposition reportedly leads to cardiac sympathetic denervation [17,18], and sympathetic denervation could be a contributing factor to SCD in patients with Parkinson’s disease [19]. However, currently, no studies have explored this aspect.

Therefore, this study examined a series of cases of sATTR-CA identified through histopathological analysis in our laboratory. We aimed to assess whether the deposition pattern of sATTR differs between the right atrium (RA) and left atrium (LA), to evaluate the relationship between ATTR and AANF deposits and to investigate whether ATTR can accumulate in the peripheral nerves of the atria.

## 2. Results

### 2.1. Clinical Profiles and Demographics

Out of 325 cases, 20 (6.2%) were sATTR amyloidosis (18.1% in individuals aged ≥80 years [17/94 cases]). Table 1 summarizes the clinical information of these patients.

Of these 20 cases, 9 were males and 11 were females (mean age: 85.6 ± 4.8 years [78–94 years] and 85.5 ± 6.0 years [79–97 years], respectively). Regarding the cause of death, 3 died from illness (with 1 related to ATTR-CA), 5 by suicide and 12 from accidental causes. Clinical history data from 19 patients revealed that 15 (79%), 5 (26%) and 3 (16%) had hypertension, hyperlipidemia and diabetes, respectively. Furthermore, 8 out of the 20 patients had a measured heart weight-to-estimated normal heart weight ratio exceeding 1.5 [20]. None of the cases had a family history of amyloidosis or had been diagnosed with ATTR-CA before death. Additionally, only one patient was clinically diagnosed with atrial fibrillation and four had coronary artery stenosis (≥75% reduction in the luminal cross-sectional area), but none had chronic heart failure prior to death.

### 2.2. Histopathological Findings of ATTR and AANF in the Heart

Table 2 summarizes the positive deposition rates of ATTR and AANF in each anatomical area.

All patients had ATTR deposition in the myocardial layer in the LA and left atrial appendage (LAA), and these areas had a significantly higher deposition frequency than the RA and right atrial appendage (RAA) (*p* = 0.02 and *p* < 0.01, respectively). Interestingly, one patient showed only vascular deposition in the ventricular septum (VS). Furthermore, ATTR deposition in the endocardial interstitium was significantly more frequent on the right side of the AS than on the left side (*p* < 0.01). Only three patients had ATTR deposition in the sinoatrial node (SAN) and none in the atrioventricular node (AVN). Neural/perineural involvement was observed in six patients, with two exhibiting both neural and perineural involvement (Figure 1).

However, in most cases, the atrial peripheral nerve fibers were free of ATTR deposition. Additionally, we incidentally found multinucleated giant cells infiltrating around ATTR deposits in two patients (patients 8 and 13; Figure 2).

The frequency of AANF deposition did not significantly differ between the LA and RA regions.

Figure 3 illustrates the semiquantitative and quantitative pathological evaluation results (all analysis results are provided in Appendix A, with a summary in Appendix A), and Figure 4 shows representative pathological microphotographs of ATTR deposition in both sides of the atrial regions.

In the VS, ATTR deposition in the myocardial interstitium was severe, whereas that in the endocardial interstitium was mild (Figure 3a). Conversely, in the AS, deposition was observed in both the myocardial and endocardial interstitium in many cases. Similar to our previous study [12], the latter was significantly more severe on the right side (Figure 3b). ATTR deposition severity in the superior vena cava (SVC) was similar to that in the AS (Figure 3c). In the semiquantitative analysis, 14 out of 20 cases showed significantly more severe ATTR deposition in the LA regions than in the RA regions (Figure 3d,e). Our quantitative analysis further supported left-side predominant ATTR deposition in the myocardium (Figure 3g). Meanwhile, although AANF deposition was generally more severe in the LA regions than in the RA regions, the difference was not statistically significant (Figure 3f). Hence, this left-side predominant deposition pattern was considered characteristic of ATTR (Figure 4).

### 2.3. Evaluation of the Colocalization of ATTR and AANF Using Single and Double IHC

In the single immunohistochemistry (IHC) results, some amyloid nodules consisted of both ATTR and AANF deposits; however, the positive areas of ATTR and AANF were observed separately, with no clear colocalization (Figure 5a–d). Thus, to elucidate whether ATTR and AANF are colocalized, we conducted double IHC on patients who showed severe ATTR deposition (grade 4) in the LA on the semiquantitative analysis. However, specimens stained with double IHC showed no clear colocalization of ATTR and AANF (Figure 5e,f).

### 2.4. Subgroups Classified by Cluster Analysis

We used cluster analysis to classify all cases into three groups according to the severity of ATTR deposition in the AS, RA, LA, RAA and LAA (Figure 6).

Table 3 lists the clinicopathological features of each group.

Among the three groups, Group 1 (eight cases) had the mildest ATTR deposition severity, thereby considered the ‘mild ATTR deposition group’. Interestingly, seven of the eight cases in this group were female; this number was significantly higher than that in Group 2.

Group 2 (nine cases) showed the most severe interstitial ATTR deposition, thereby considered the ‘severe interstitial ATTR deposition group’. In contrast to Group 1, seven out of nine cases were male; thus, ATTR deposition in the myocardium may be more severe in males.

In Group 3 (three cases), the severity of ATTR deposition in the interstitium was not significantly different from Group 1, but it had the most severe vascular deposition. Thus, Group 3 was considered the ‘vessel-predominant ATTR deposition group’.

Compared with ATTR deposition, AANF deposition severity did not significantly differ among the three groups.

### 2.5. Comparison of ATTR and AANF Deposition Severity between Males and Females

Table 4 summarises the results of the semiquantitative and quantitative analyses for females and males. 

As shown by the cluster analysis, males were more severely affected than females in almost all measures of ATTR deposition, with statistically significant differences observed in the semiquantitative analysis of the VS interstitium and the quantitative analysis of the LA interstitium. Regarding AANF deposition severity, we noted no differences between the sexes.

## 3. Discussion

This study showed that sATTR-CA predominantly involves the LA regions. LA function evaluation has been widely reported to be useful for discriminating CA from other diseases with LV hypertrophy and predicting prognosis in patients with CA [11,21,22,23,24]. Thus, our study results provide histopathological support for these clinical observations. To our knowledge, this research is the first autopsy-based study demonstrating left-predominant ATTR deposition patterns in the atria. Additionally, two patients (patients 10 and 19) exhibiting severe ATTR deposition in the RA also had severe ATTR deposition in the VS and LA; hence, RA involvement may be the most advanced phase in sATTR-CA. Consistent with this finding, Singulane et al. reported that abnormalities in RA size and strain are associated with a worse prognosis in patients with CA [9]. By comparing the amount of deposition in the LA and RA regions, ATTR-CA may be early detected. Differences in ATTR-CA in both atria are currently unreported and require further investigation. Furthermore, Palmer et al. reported that LA function is more impaired in ATTR-CA cases than in AL-CA cases [25]. By characterizing amyloid deposition in the left and right atria histopathologically in AL-CA cases, the differentiation of the two diseases may be more accurate on imaging examination.

Our previous study using the proximity ligation assay demonstrated the colocalization of ANF and TTR in amyloid deposits in the AS [12]. Furthermore, Bandera et al. found ANF in the atrial ATTR deposits in three of five patients through a proteomic analysis [11]. In addition, AANF deposition is reportedly more severe in the LA than in the RA regions [13,14,15]. Taken together, the varying severity of ATTR deposition in the left and right sides of the AS and atria might be explained by the presence of synergistic effects between AANF and ATTR; however, we could not demonstrate this hypothesis through IHC methods. Regarding AANF, immunoreactivity tended to weaken as congophilia became stronger, making the evaluation through IHC difficult. This limitation is probably caused by the fact that the epitope recognized by the antibody used in this study was hidden as amyloid fibril formation progressed [26]; the same phenomenon may have prevented the detection of AANF deposits when combined with ATTR deposits. In histopathology specimens, proving whether AANF is codeposited with ATTR or whether non-amyloid-forming ANF is merely present in the ATTR deposits is difficult. Verifying the existence of cross-seeding is also nearly impossible. Thus, drawing conclusions from morphological studies alone is challenging, and in vitro investigations to evaluate cross-seeding activity are necessary to clarify the relationship between the two [27].

In addition to the presence of a synergetic interaction between ATTR and AANF, we have three other hypotheses for the cause of the left-side predominant ATTR deposition pattern in the atria. The first hypothesis is mechanical stimulation. Repetitive mechanical stimulus could favor ATTR fibrogenesis and/or tissue infiltration [28,29]. Given that the mean pressure is higher in the LA than in the RA, the former might have stronger mechanical stress than the latter, resulting in left-predominant ATTR deposition. Second, the RA has resistance to ATTR deposition. As shown in our study, ATTR deposition is less likely to occur in conduction system tissues [12]. Therefore, ATTR deposition is less likely to occur in the RA, where the cardiac conduction system is concentrated, than in the LA, where it is absent. Third, a mechanism that can remove ATTR deposition exists. Some amyloid deposits cause granulomatous inflammation with multinucleated cells, probably as a result of an immune response to them [30,31]. Regarding ATTR deposits, we recently reported that granulomas associated with sarcoidosis break ATTR deposits, indicating that this inflammation type is necessary for amyloid removal [32]. Moreover, Fontana et al. reported three cases of antibody-induced reversal of ATTR amyloidosis-related cardiomyopathy, one of which underwent myocardial biopsy and showed multinucleated giant cell infiltration around ATTR deposits [33]. The present study found two cases exhibiting similar findings. In these cases, the RA region had no interstitial ATTR deposition, whereas the LA region had grade 2 or higher interstitial ATTR deposition, suggesting that the right side is more capable of removing deposits. This resistance to amyloid deposition in the RA system may also be associated with left-predominant AANF deposition [13,14,15]. These hypotheses, if proven, could contribute to the development of not only therapeutic but also preventive strategies against ATTR-CA and should be explored in the future.

Our cluster analysis revealed that the burden of ATTR deposition in the myocardium, especially in the VS and LA, is significantly more severe in males than in females. As previously reported, sex differences appear to be closely related to ATTR-CA severity [12,34,35,36]. Sex hormone involvement has been suggested as one of the causes of this difference [34,37]. Interestingly, an association with sex hormones has also been noted for amyloid-beta [38,39], one of the common age-related amyloid deposits in older adults [40]. Thus, the role of sex hormones in the pathogenesis of age-related amyloidosis should be given more attention.

Sensory neuropathy reportedly occurs in 28.4% of patients with wild-type ATTR amyloidosis, suggesting that it can cause peripheral nervous system damage [41,42]. Consistent with this finding, our study showed that sATTR amyloidosis can extend into the epineurium and nerve fibers of peripheral nerves of the heart. To our knowledge, this study is the first to histopathologically demonstrate that ATTR deposition can extend to atrial peripheral nerves, suggesting that it can be one of the causes of SCD in sATTR-CA cases. However, epineurium/neural involvement was found only in cases showing grade 3 or higher myocardial interstitial infiltration and only to a very mild degree. In line with this finding, Gimelli et al. reported that in patients with wild-type ATTR, cardiac sympathetic denervation tended to worsen in parallel with the amyloid burden, although their correlation did not achieve statistical significance [17]. Taken together, myocardial denervation in sATTR-CA may not occur until at an advanced stage, and it could be milder than that in hereditary ATTR amyloidosis [18]. The significance of peripheral neuropathy in sATTR amyloidosis, including wild-type ATTR amyloidosis, needs further investigation.

In addition to a certain level of bias in our study population, this study was limited by clinical information, including the presence or absence of cardiac symptoms, carpal tunnel syndrome, or spinal stenosis in some patients, mainly because of the lack of severe clinical symptoms or low cardiologist consultation rates. Moreover, a small sample size (only 20 cases) was another limitation of this study. The validity of this study would be better demonstrated with a larger sample size and a more diverse population. Additionally, we could not test the *TTR* gene and excluded cases of geriatric onset of hereditary ATTR amyloidosis [43]. Thus, we described the cases as ‘sporadic ATTR’ instead of ‘wild-type ATTR’. The selection of ATTR deposition-positive cases was performed by one cardiovascular pathologist (N.N.) based on the histopathological and IHC findings, and the semiquantitative ATTR and AANF deposition burden analysis was performed separately by another cardiovascular pathologist (S.I.). Thus, in the latter processes, all assessments were performed blinded to clinical information and pathological diagnoses.

In conclusion, this study demonstrated a left-dominant deposition pattern of ATTR in the atrial regions. We previously reported that the severity of ATTR deposition differed between the right and left atrial septal endocardia [12]. Herein, we demonstrated significant differences in the amount of ATTR deposition in the atrial myocardium, which is most involved in atrial function. Therefore, the results of the present study may be more directly related to atrial function than previous findings and are likely to be clinically important. Together with the presence of a group of cases showing atrial predominance of deposition [12], ATTR deposition might occur irregularly in the heart between both atria and between atria and ventricles. Regarding AANF deposition, it might occur in the atria separately from ATTR deposition, accentuating the irregularity of atrial amyloid deposition. Thus, these deposits could induce arrhythmias by irregularly disrupting the stimulatory conduction system within the atria and from the atria to the ventricles. Moreover, this disruption might be further aggravated in advanced stages by cardiac innervation impairment. These findings underscore the importance of atrial evaluation in sATTR-CA cases to detect early and differentiate from other conditions that cause cardiac hypertrophy. In particular, differentiating ATTR-CA from AL-CA is crucial for determining the treatment strategy; thus, findings that help differentiate between the two are needed. Currently, no detailed histopathological studies have reported on the left-right differences in the deposition pattern of AL-CA in the atria. Hence, future studies are necessary.

## 4. Materials and Methods

### 4.1. Subjects

In our department, we retrospectively examined autopsy records documented from July 2019 to July 2021, during which samples from the LA, RA, LAA and RAA were routinely collected. Using these records, we evaluated heart specimens from 325 serial autopsy subjects in whom all organs, including the brain, could be sampled. Patients’ demographic and clinical characteristics (including the cause of death) were retrieved from the medical records of police examinations and contributions from family members or from the primary physician if a record indicated clinic visits. This study obtained approval from the Ethical Committee of Toyama University (I2020006) and conformed to the ethical standards outlined in the 1964 Declaration of Helsinki and its 2008 amendment.

### 4.2. Tissue Samples

We sampled one block (containing 2–4 pieces of tissue) from each atrium and atrial appendage. The amount of amyloid deposition was evaluated in both atrial regions, AS, basilar VS at the AVN level and SVC at the SAN level. The specimens were fixed in 20% buffered formalin and routinely embedded in paraffin. Then, they were sectioned at 4 μm thick, followed by hematoxylin and eosin staining, elastica–Masson staining, or IHC. We also prepared 6 μm thick sections for phenol Congo red (pCR) staining [44].

### 4.3. Semiquantitative Grading System for ATTR and AANF Deposition

CA was screened according to a previously described method in our department [12]. Following confirmation of ATTR and AANF deposition by pCR staining and IHC, the severity of ATTR and AANF deposition in the myocardial interstitium, vessels and endocardial interstitium was assessed semiquantitatively. For this evaluation, we used a 5-point scoring system, taking into account the deposition patterns described in the existing literature [3,13,45] as follows:

ATTR interstitial grading:Grade 0: None.Grade 1: Focal and mild deposition in the perimyocyte area, without nodular lesions.Grade 2: Multifocal deposition in the perimyocyte area, with minimal nodular lesions replacing the normal tissue.Grade 3: Multifocal deposition, with some nodular lesions.Grade 4: Diffuse deposition, with some nodular lesions.

ATTR vascular grading:Grade 0: None.Grade 1: Focal deposition.Grade 2: Multifocal deposition with minimal involvement of the full thickness of the wall.Grade 3: Multifocal deposition, with some occupying the full thickness of the wall.Grade 4: Multifocal deposition, exhibiting an obstructive change (≥75% reduction in the luminal cross-sectional area).

ATTR endocardium grading:Grade 0: None.Grade 1: Focal and mild deposition in the subendocardial fibrous tissue.Grade 2: Multifocal deposition with minimal massive deposits.Grade 3: Multifocal deposition with some massive deposits.Grade 4: Multifocal deposition with several massive deposits, with some occupying the full thickness of the endocardial fibrous tissue.

AANF deposition grading:Grade 0: None.Grade 1: Focal and mild deposition in the interstitium and/or vessels.Grade 2: Multifocal deposition in the interstitium and vessels, with some surrounding the entire cardiomyocyte circumference.Grade 3: Diffuse deposition surrounding the entire cardiomyocyte circumference.Grade 4: Diffuse deposition with some nodular deposits.

Figure 7 shows representative microphotographs illustrating the deposition patterns of this grading system. Figure 8, Figure 9 and Figure 10 provide representative microphotographs of AANF deposition in the atrium and AS.

ANF immunoreactivity was detected not only in AANF deposits but also in the cytoplasm of cardiomyocytes, vessels and endocardium, sometimes exhibiting a disparity with pCR staining intensity (Figure 8, Figure 9 and Figure 10). Consequently, the assessment of AANF deposition severity focused solely on the deposition pattern around cardiomyocytes in pCR-stained specimens, utilizing AANF IHC to exclude ATTR deposition. All histopathological gradings were determined at the site with the most amyloid deposits observed under 100× magnification. Additionally, we assessed whether AVN, SAN, neural and perineural (epineurium) involvement existed in the atrial peripheral nerves.

### 4.4. Quantitative Analysis of ATTR Deposition Burden

We quantitatively assessed the amount of ATTR deposition burden in the myocardium in the VS, AS, both atria and atrial appendages, by using Image J/Fiji (version 1.54f) [46] as previously described [47,48]. Using a bright field microscope under 100× magnification (area per image: 1.68 × 1.32 mm), we photographed one location with the most severe ATTR deposition and then calculated the ATTR burden value. The microphotographs were acquired using a BX51 microscope (Olympus Corporation, Tokyo, Japan) equipped with a DP27 microscope camera and CellSens Standard imaging software (version 3.1, Olympus Corporation). All microphotographs were separately imported to Photoshop (Adobe Inc., San Jose, CA, USA) and preprocessed by adjusting the contrast to recognize small deposits clearly and reduce the background noise of nonspecific immunoreactivity [49]. To improve measurement accuracy, we only assessed regions of 100 pixels or larger in area, considering the tendency to erroneously recognize lipofuscin in myocytes in patients with low ATTR deposition. In the quantitative analysis, cases of negative deposition in the myocardial interstitium in the semiquantitative analysis were counted as 0.

### 4.5. Single and Double IHC

Single IHC was performed using primary antibodies against prealbumin (transthyretin) (rabbit, clone EPR3219, 1:2000, Abcam, Cambridge, UK) and ANF (mouse, clone 23/1, 1:20,000; GeneTex, Irvine, CA, USA). Antigens were retrieved using 98% formic acid for 1 min. In this IHC, we used the Leica Bond-MAX automation system and Leica Refine detection kits (Leica Biosystems, Nussloch, Germany) according to the manufacturer’s instructions. All sections were counterstained with hematoxylin.

In double IHC, the same antibodies mentioned above were used to assess whether ATTR and AANF were colocalized. First, IHC for AANF was conducted using the same procedure as that for single IHC. After the initial IHC, sections were incubated with 0.3% H_2_O_2_ for 10 min, followed by primary antibodies against transthyretin (overnight at 4 °C). Signal development was achieved using the immunoenzyme polymer method (Histofine Simple Stain MAX PO Multi; Nichirei Biosciences, Tokyo, Japan) with the Vina Green Chromogen Kit (BioCore Medical Technologies, Silver Spring, MD, USA) for 5 min. We counterstained all sections with hematoxylin.

### 4.6. Statistical Analysis

Data were analyzed using IBM SPSS statistics version 29 (SPSS Inc., Chicago, IL, USA). Significance levels were set at *p* < 0.05 utilizing a two-tailed approach. Fisher’s exact test was used for categorical variables (pathological findings). Ordinal variables (pathological scores) were compared using the Mann–Whitney U test (comparison between two groups) or the Kruskal–Wallis test with post-Bonferroni correction (comparison of three or more groups). Ward’s hierarchical cluster analysis was performed using the histopathological score described above to classify the pathologically defined cardiac ATTR cases into subgroups.

## Figures and Tables

**Figure 1 ijms-25-08176-f001:**
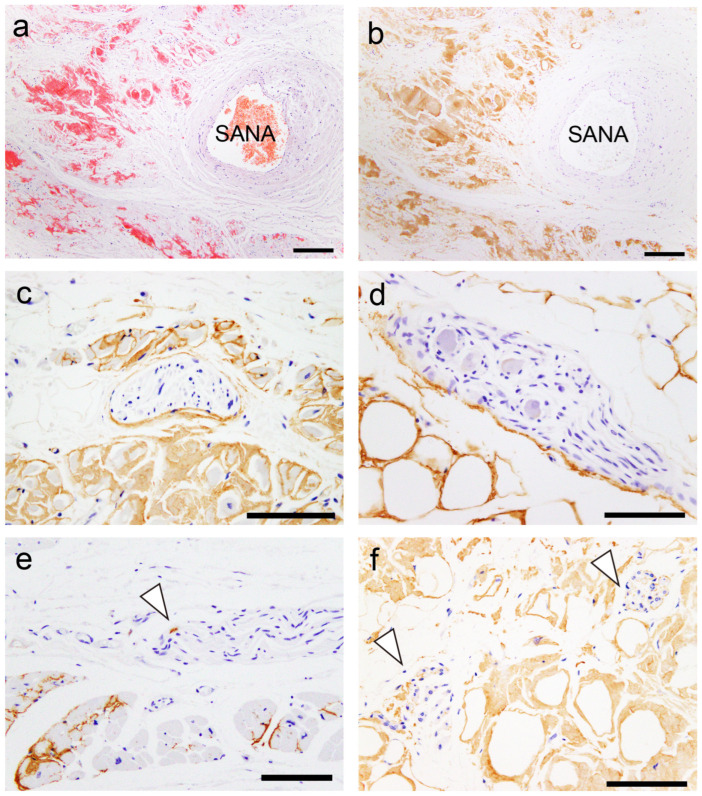
Representative microphotographs of the sinoatrial node (SAN), perineural (epineurium) and neural involvement. (**a**,**b**) SAN. (**c**–**f**) Atrium. (**a**) Phenol Congo red (pCR) staining. (**b**–**f**) Immunohistochemistry for transthyretin. (**a**,**b**) Transthyretin-derived amyloid deposition in the SAN (SANA, sinoatrial node artery). Perineural (epineurium) (**c**) and peri-ganglion involvement. (**e**,**f**) Neural involvement (arrowheads). Scale bar = 200 μm (**a**,**b**); 100 μm (**c**–**f**).

**Figure 2 ijms-25-08176-f002:**
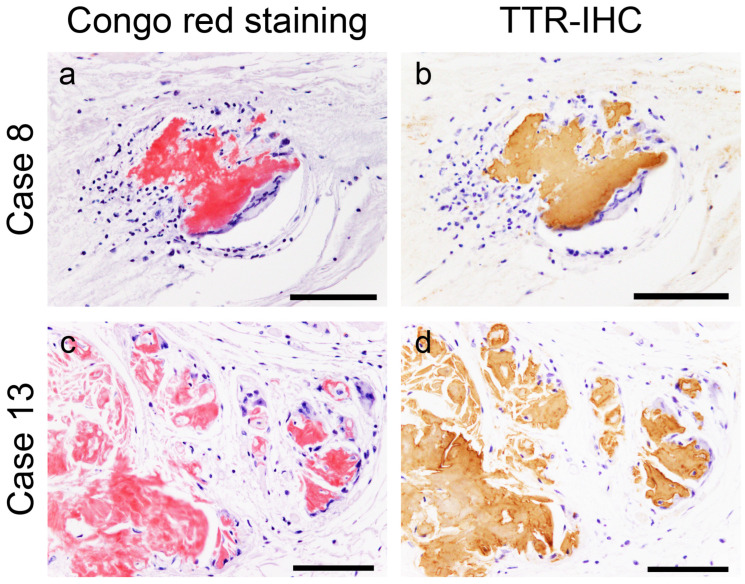
Representative microphotographs of multinucleated giant cells surrounding transthyretin-derived amyloid (ATTR) deposits. (**a**,**b**) Atrial septum; (**c**,**d**) left atrial appendage. (**a**,**c**) pCR staining; (**b**,**d**) immunohistochemistry (IHC) for transthyretin. Multinucleated giant cells with lymphocytes surround ATTR deposits. No clear fragmentation of ATTR deposits is observed. Scale bar = 100 μm (**a**–**d**).

**Figure 3 ijms-25-08176-f003:**
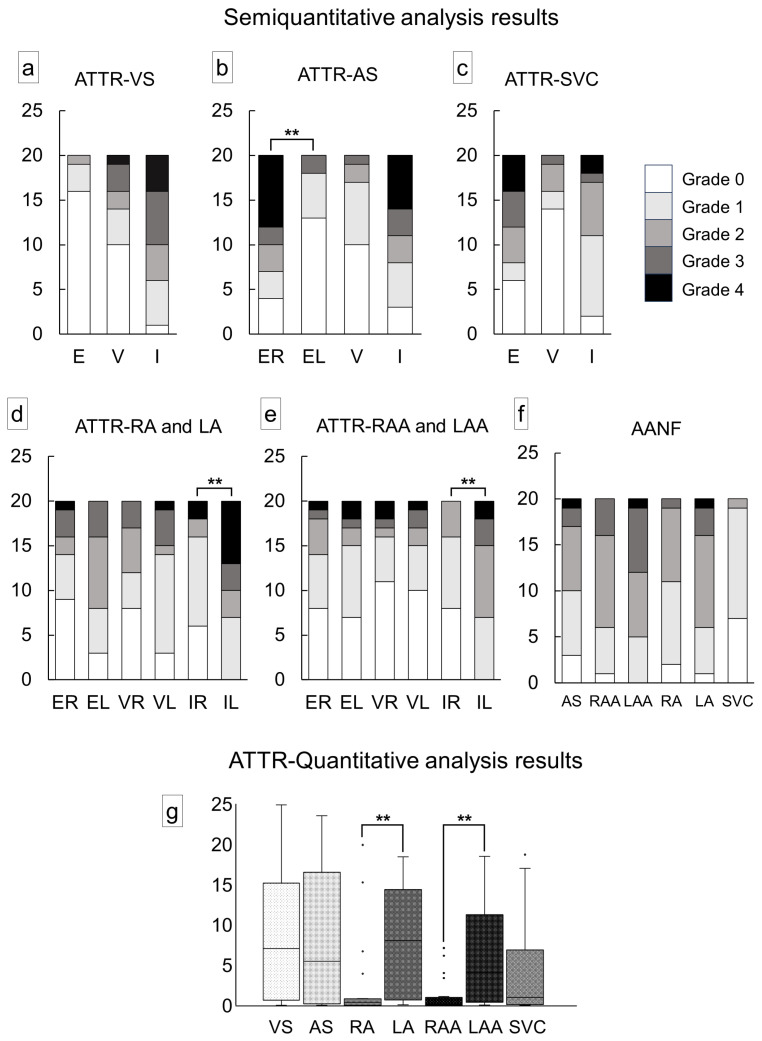
Severity of cardiac involvement in each region of the heart. (**a**–**e**) Semiquantitative grading of ATTR deposition; (**f**) semiquantitative grading of AANF deposition; (**g**) quantitative grading of ATTR deposition in the myocardium. (**a**) Ventricular septum (VS); (**b**) atrial septum (AS); (**c**) superior vena cava (SVC); (**d**) right atrium (RA) and left atrium (LA); (**e**) right atrial appendage (RAA) and left atrial appendage (LAA); (**f**) AS, both atrial regions and SVC; (**g**) VS, AS, both atrial regions and SVC. Abbreviations: E, endocardium; EL, endocardium of the left side; ER, endocardium of the right side; I, interstitium; IL, interstitium of the left side; IR, interstitium of the right side; V, vessel; VL, vessel of the left side; VR, vessel of the right side. ** *p* < 0.01, compared using the Mann–Whitney U test.

**Figure 4 ijms-25-08176-f004:**
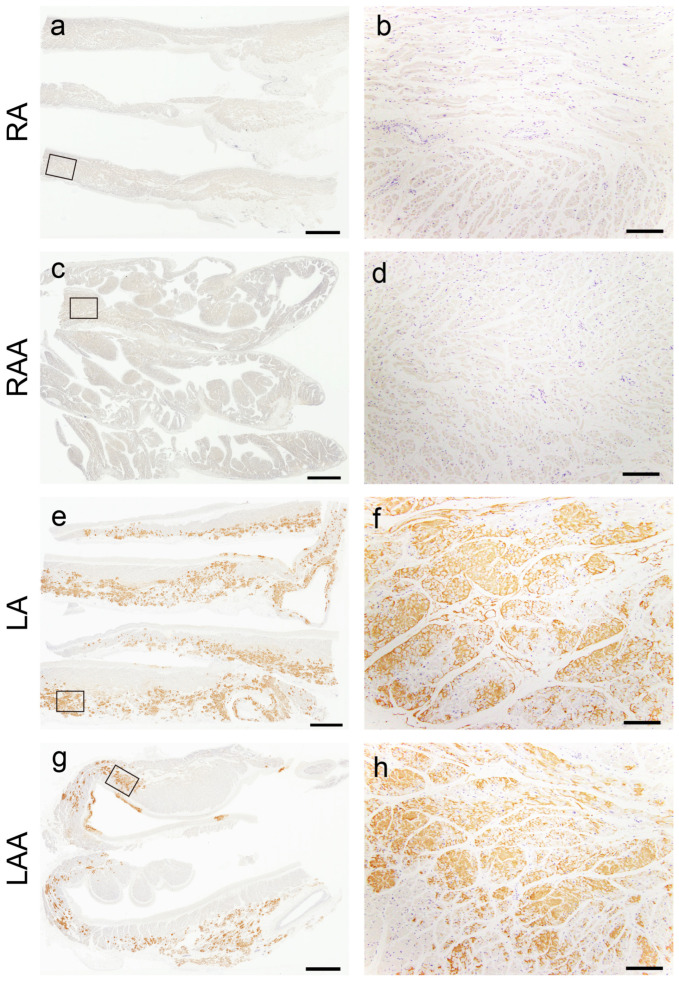
Representative microphotographs of the left-side predominant deposition pattern of ATTR in the atrial regions. (**a**–**h**) Transthyretin IHC (patient 8). (**a**,**b**) Right atrium; (**c**,**d**) right atrial appendage; (**e**,**f**) left atrium; (**g**,**h**) left atrial appendage. Panels (**b**), (**d**), (**f**) and (**h**) are a higher magnification view indicated with a black square in panels (**a**), (**c**), (**e**) and (**g**), respectively. ATTR was not deposited in the right atrial regions (**a**–**d**) but was severely deposited in the left atrial regions (**e**–**h**). Scale bar = 3 mm (**a**,**c**,**e**,**g**); 200 μm (**b**,**d**,**f**,**h**).

**Figure 5 ijms-25-08176-f005:**
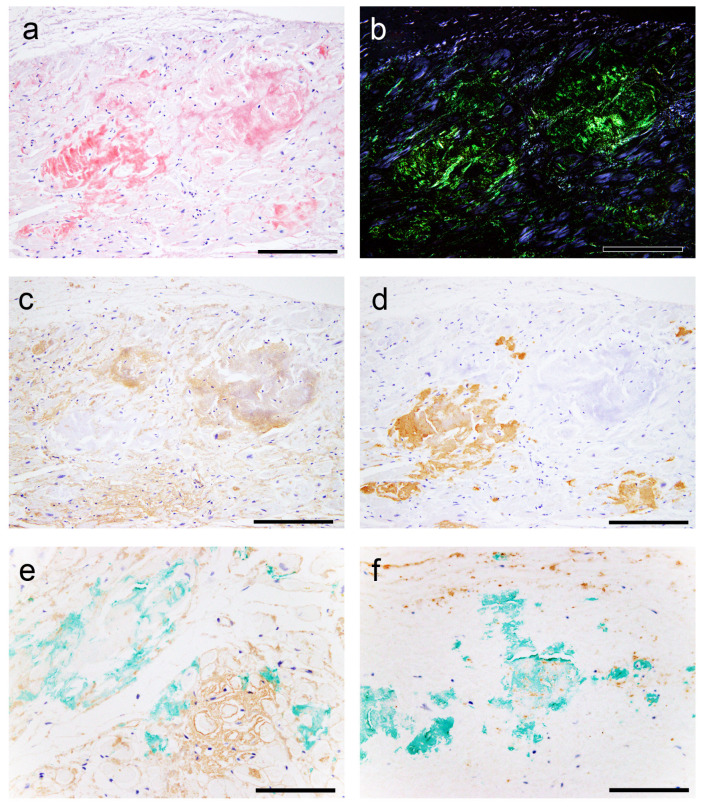
Representative microphotographs of nodular deposits in the LA evaluated using double IHC for ANF and transthyretin. (**a**–**f**) Patient 10. (**a**,**b**) pCR staining under bright field (**a**) and polarized light (**b**). Single IHC for ANF (**c**) and transthyretin (**d**). Double IHC for ANF and transthyretin (**e**,**f**). (**a**–**d**) Nodular lesions observed in the right half are positive for ANF but negative for transthyretin, whereas those observed in the left half are negative for ANF but positive for transthyretin. (**e**,**f**) Double IHC shows no clear colocalization of the two (ANF, brown; transthyretin, green) in the myocardial layer (**e**) or endocardial interstitium (**f**). Scale bar = 100 μm (**a**–**f**).

**Figure 6 ijms-25-08176-f006:**
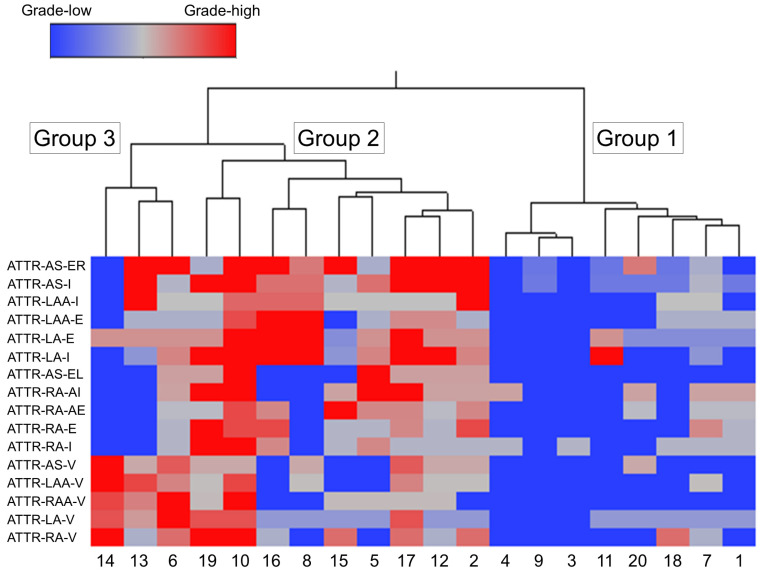
Wards hierarchical cluster analysis using the semiquantitative ATTR deposition grading. Dendrogram of cluster analysis (upper panel) and amyloid deposition distribution and severity in each area (lower panel). Abbreviations: E, endocardium; EL, endocardium of the left side; ER, endocardium of the right side; I, interstitium; V, vessel.

**Figure 7 ijms-25-08176-f007:**
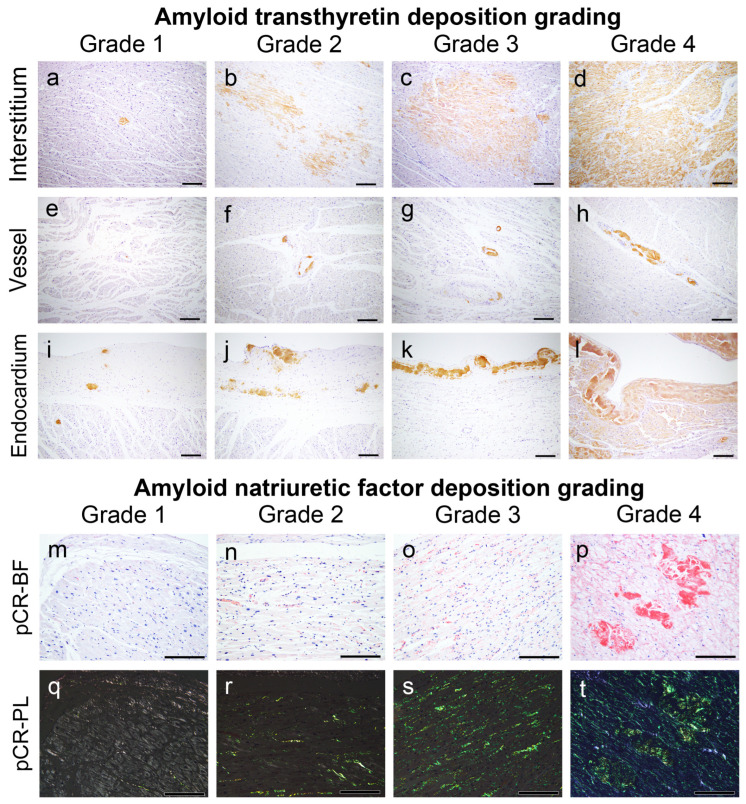
Representative microphotographs of ATTR and AANF deposition grading employed in this study. (**a**–**l**) IHC for transthyretin; (**m**–**t**) pCR staining under bright field (**m**–**p**) and polarized light observation (**q**–**t**). Scale bar = 200 μm (**a**–**t**).

**Figure 8 ijms-25-08176-f008:**
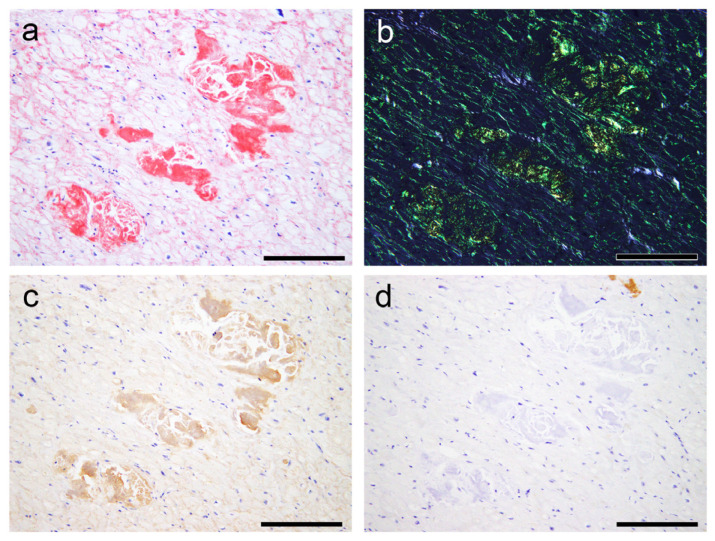
Representative microphotographs of AANF deposition with nodular deposits. (**a**–**d**) Left atrium (Case 7). (**a**,**b**) pCR staining under bright field (**a**) and polarized light (**b**) observation; IHC for ANF (**c**) and transthyretin (**d**). (**a**) Both nodular and pericellular deposition patterns are observed. (**b**) Both lesions show clear apple-green birefringence under polarized light. However, although nodular deposits exhibit moderate to strong immunoreactivity for ANF, pericellular deposits show very weak immunoreactivity. (**d**) These deposits are negative for transthyretin. Scale bar = 200 μm (**a**–**d**).

**Figure 9 ijms-25-08176-f009:**
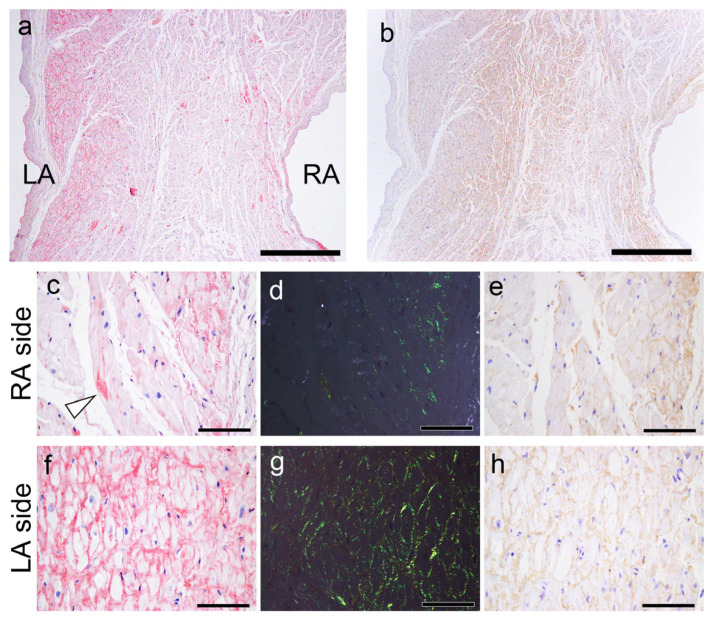
Representative microphotographs of AANF deposition in the AS. (**a**,**c**,**d**,**f**,**g**) pCR staining under bright field (**a**,**c**,**f**) and polarized light (**d**,**g**) observation; IHC for ANF (**b**,**e**,**h**). (**a**) CR-positive deposition is observed predominantly on the LA side compared with the RA side. Note that AANF deposition is predominantly observed in the subendocardial myocardial layer on both the left and right sides. (**b**) However, in the ANF-IHC specimen, the difference is not obvious. In a higher magnification view, on the RA side, AANF deposits are observed that incompletely surround the cardiomyocytes (arrowhead indicates ATTR deposits) (**c**–**e**). In contrast, on the LA side, AANF deposits are observed that completely surround the cardiomyocytes (**f**–**h**). Nevertheless, note that the immunoreactivity of ANF on the RA side (**e**) is equal to or slightly weaker than that on the LA side (**h**). Scale bar = 1 mm (**a**,**b**); 100 μm (**c**–**h**).

**Figure 10 ijms-25-08176-f010:**
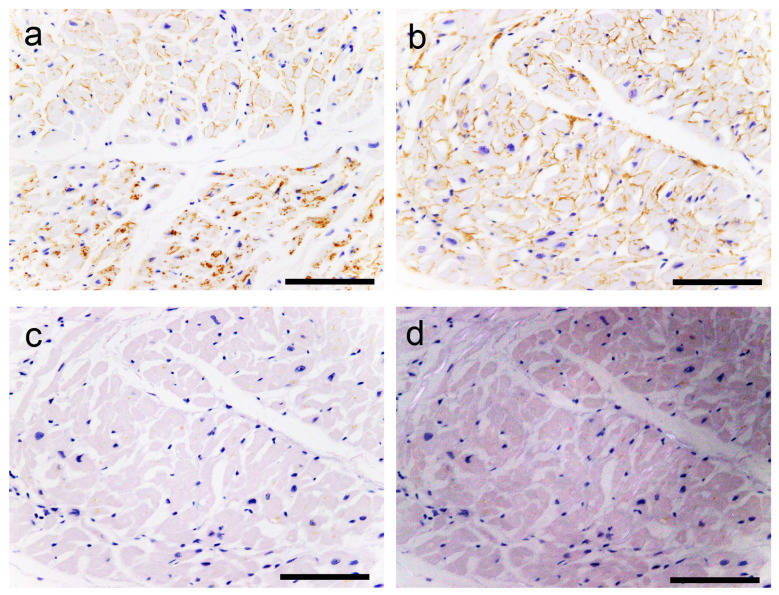
Representative microphotographs of AANF deposition without congophilia. (**a**–**d**) LA. (**a**,**b**) IHC for ANF; (**c**,**d**) pCR staining under bright field (**c**) and polarized light (**d**) observation. (**a**) The upper half of the panel (the endocardium side) shows ANF deposition around cardiomyocytes. In contrast, in the lower half, ANF immunoreactivity is observed in the cytoplasm of the cardiomyocytes. (**b**) Strong ANF immunoreactivity is observed around cardiomyocytes. (**c**,**d**) However, these deposits are negative for pCR staining and show no birefringence. Scale bar = 100 μm (**a**–**d**).

**Table 1 ijms-25-08176-t001:** Summary of the clinical and pathological features of ATTR-CA cases.

#	Age	Sex	BH	BW	BMI	HW	NHW *	Ratio **	C/TD	HT	HL	DM	ART	CHF	CAS
1	81	F	147	43.6	20.2	346	242	1.4	HyT/Sui	Neg	Pos	Neg	Neg	Neg	Rt
2	87	F	147	36.3	16.8	321	218	1.5	Dro/Sui	Pos	Neg	Neg	Neg	Neg	Neg
3	80	F	146	45.9	21.5	387	248	1.6	HyT/Acc	Pos	Neg	Pos	Neg	Neg	Neg
4	79	F	164	72	26.8	584	361	1.6	Dro/Acc	Pos	Neg	Neg	Af	Neg	Neg
5	82	M	160	53.1	20.7	556	317	1.8	HyT/Ill	Pos	Pos	Neg	Neg	Neg	Neg
6	90	F	151	39.1	17.1	336	234	1.4	HyT/Acc	Neg	Neg	Neg	Neg	Neg	Neg
7	89	M	158	40.4	16.2	330	265	1.2	Dro/Acc	Pos	Neg	Neg	Neg	Neg	Neg
8	78	M	156	65.3	26.8	550	350	1.6	CA/Ill	Pos	Neg	Pos	Neg	Neg	Neg
9	90	F	148	44.7	20.4	306	247	1.2	Dro/Acc	Pos	Neg	Neg	Neg	Neg	Neg
10	97	F	153	51.5	22.0	356	278	1.3	HS/Acc	Neg	Neg	Neg	Neg	Neg	Neg
11	79	F	153	49.8	21.3	302	272	1.1	Burn/Acc	Pos	Pos	Neg	Neg	Neg	Neg
12	83	M	158	41.5	16.6	383	269	1.4	Dro/Sui	Pos	Neg	Neg	Neg	Neg	Neg
13	85	F	145	35.2	16.7	283	211	1.3	Burn/Acc	NA	NA	NA	NA	NA	Lt
14	94	M	159	29.4	11.6	379	219	1.7	Dro/Acc	Neg	Neg	Neg	Neg	Neg	Neg
15	81	M	161	58.5	22.6	450	339	1.3	Dro/Acc	Pos	Pos	Neg	Neg	Neg	Neg
16	85	M	157	51.8	21.0	364	306	1.2	Dro/Sui	Pos	Neg	Neg	Neg	Neg	Neg
17	89	M	158	44.2	17.7	304	280	1.1	HyT/Acc	Pos	Neg	Neg	Neg	Neg	Lt
18	80	F	152	69.6	30.1	408	329	1.2	Dro/Acc	Pos	Neg	Neg	Neg	Neg	Neg
19	89	M	159	37.4	14.8	435	254	1.7	DOD/Sui	Pos	Neg	Pos	Neg	Neg	Neg
20	93	F	144	41.8	20.2	365	232	1.6	ACD/Ill	Pos	Pos	Neg	Neg	Neg	Rt, Lt

Abbreviations: Acc, accident; ACD, acute cardiac dysfunction (excluding CA); Af, atrial fibrillation; ART, arrhythmia; BH, body height (cm); BW, body weight (kg); BMI, body mass index; CAS, coronary artery stenosis (≥75%); CHF, chronic heart failure; DM, diabetes mellitus; F, female; HL, hyperlipidemia; HT, hypertension; HW, heart weight; Ill, illness; Lt, left; M, male; NA, not available; NHW, normalized heart weight (g); Rt, right; Sui, suicide. * Values were calculated with reference to the literature [20]. ** HW/NHW.

**Table 2 ijms-25-08176-t002:** Summary of the positive deposition rates of ATTR and AANF in each area.

ATTR Deposition Rate (%)	Total # of Cases = 20
VS (E/V/I)	4 (20)/10 (50)/19 (95)
AS (ER/EL/V/I)	16 (80) **/7 (35)/10 (50)/17 (85)
RA (E/V/I)	11 (55)/12 (60)/14 (70)
RAA (E/V/I)	12 (60)/9 (45)/12 (60)
LA (E/V/I)	17 (85)/17 (85)/20 (100) *
LAA (E/V/I)	14 (70)/10 (50)/20 (100) **
SVC (E/V/I)	14 (70)/6 (30)/18 (90)
Sinoatrial node	3 (15)
Atrioventricular node	0
Neural/perineural deposition	2 (10)/6 (30)
AANF deposition rate (%)	Total # of cases = 20
AS	17 (85)
RA	18 (90)
RAA	19 (95)
LA	19 (95)
LAA	20 (100)
SVC	13 (65)

Abbreviations: AS, atrial septum; E, endocardium; EL, endocardium in the left side; ER, endocardium in the right side; I, interstitium; LA, left atrium; LAA, left atrial appendage; RA, right atrium; RAA, right atrial appendage; SVC, superior vena cava; V, vessel; VS, ventricular septum. * *p* < 0.05; ** *p* < 0.01, right side vs. left side, compared using the Fisher’s exact test.

**Table 3 ijms-25-08176-t003:** Summary of clinical and histopathological characteristics among three groups classified by cluster analysis.

		Group 1 (*n* = 8)	Group 2 (*n* = 9)	Group 3 (*n* = 3)	*p* Value *
Sex (F/M)		7/1	2/7	2/1	**0.02**/0.49/0.24
Age (range)		83.9 ± 5.4 (79–93)	85.7 ± 5.3 (78–97)	89.7 ± 3.7 (85–94)	0.24 (1.00/0.28/0.67)
N/PN involvement		1 (13)	4 (44)	1 (33)	0.29/0.49/1
ATTR deposition severity					
VS	E	0	0.6 ± 0.7 (0–2)	0	0.06 (0.08/1/0.32)
	V	0.3 ± 0.4 (0–1)	1.1 ± 1.2 (0–3)	3.0 ± 0.8 (2–4)	**0.02** (0.50/**0.01**/0.18)
	I-SQ	1.6 ± 0.7 (1–3)	3.3 ± 0.7 (2–4)	1.3 ± 1.2 (0–3)	**<0.01** (**<0.01**/1/0.06)
	I-Q	2.9 ± 3.1 (0–9.2)	15.6 ± 7.4 (2.3–24.9)	3.2 ± 4.5 (0–9.6)	**<0.01** (**<0.01**/1/**0.03**)
AS	ER	1.0 ± 1.0 (0–3)	3.4 ± 0.8 (2–4)	2.7 ± 1.9 (0–4)	**<0.01** (**<0.01**/0.34/1)
	EL	0	1.1 ± 1.1 (0–3)	0.3 ± 0.5 (0–1)	**0.02** (**0.02**/1/0.77)
	V	0.1 ± 0.3 (0–1)	0.8 ± 0.6 (0–2)	2.0 ± 0.8 (1–3)	**<0.01** (0.16/**<0.01**/0.32)
	I-SQ	0.9 ± 0.6 (0–2)	3.4 ± 0.7 (2–4)	2.0 ± 1.6 (0–4)	**<0.01** (**<0.01**/0.80/0.50)
	I-Q	0.6 ± 0.8 (0–2.4)	14.7 ± 5.2 (6.1–23.5)	8.6 ± 8.9 (0–20.8)	**<0.01** (**<0.01**/0.65/0.68)
RA	E	0.4 ± 0.7 (0–2)	2.0 ± 1.2 (0–4)	0.3 ± 0.5 (0–1)	**0.02** (**0.02**/1/0.16)
	V	0.4 ± 0.7 (0–2)	1.6 ± 1.1 (0–3)	2.0 ± 0.8 (1–3)	**0.03** (0.09/0.09/1)
	I-SQ	0.6 ± 0.5 (0–1)	1.8 ± 1.3 (0–4)	0.3 ± 0.5 (0–1)	**0.047** (0.15/1/0.11)
	I-Q	0.2 ± 0.3 (0–0.7)	5.3 ± 6.9 (0–19.9)	0.2 ± 0.3 (0–0.7)	**0.04** (0.07/1/0.20)
RAA	E	0.4 ± 0.5 (0–1)	1.9 ± 1.1 (0–4)	0.3 ± 0.5 (0–1)	**0.01** (**0.02**/1/0.11)
	V	0	1.0 ± 1.2 (0–4)	3.0 ± 0.8 (2–4)	**<0.01** (0.07/**<0.01**/0.24)
	I-SQ	0.5 ± 0.5 (0–1)	1.2 ± 0.8 (0–2)	0.3 ± 0.5 (0–1)	0.10 (0.20/1/0.27)
	I-Q	0.1 ± 0.1 (0–0.2)	2.5 ± 2.6 (0–7.1)	0.0 ± 0.1 (0–0.1)	**0.03** (0.053/1/0.18)
LA	E	0.8 ± 0.7 (0–2)	2.3 ± 0.7 (1–3)	2.0 ± 0.0 (2)	**<0.01** (**<0.01**/0.18/1)
	V	0.6 ± 0.5 (0–1)	1.7 ± 0.9 (1–3)	3.0 ± 0.8 (2–4)	**<0.01** (0.11/**<0.01**/0.34)
	I-SQ	1.5 ± 1.0 (1–4)	3.6 ± 0.7 (2–4)	2.0 ± 0.8 (1–3)	**<0.01** (**<0.01**/1/0.23)
	I-Q	2.4 ± 4.3 (0.1–13.8)	11.9 ± 5.9 (1.7–18.4)	5.4 ± 3.8 (0.1–9.0)	**<0.01** (**<0.01**/1/0.27)
LAA	E	0.4 ± 0.5 (0–1)	2.0 ± 1.3 (0–4)	0.7 ± 0.5 (0–1)	**0.02** (**0.02**/1/0.47)
	V	0.1 ± 0.3 (0–1)	1.0 ± 0.9 (0–3)	3.0 ± 0.8 (2–4)	**<0.01** (0.17/**<0.01**/0.19)
	I-SQ	1.3 ± 0.4 (1–2)	2.6 ± 0.7 (2–4)	2.3 ± 1.2 (1–4)	**<0.01** (**<0.01**/0.38/1)
	I-Q	1.2 ± 1.4 (0–3.6)	9.5 ± 5.8 (1.8–18.5)	7.4 ± 6.8 (0.1–16.5)	**<0.01** (**<0.01**/0.44/1)
SVC	E	0.9 ± 1.1 (0–3)	2.7 ± 1.2 (0–4)	2.3 ± 1.7 (0–4)	0.06 (0.06/0.50/1)
	V	0	0.6 ± 0.8 (0–2)	2.0 ± 0.8 (1–3)	**<0.01** (0.50/**<0.01**/0.08)
	I-SQ	0.9 ± 0.3 (0–1)	2.1 ± 1.0 (1–4)	2.3 ± 1.7 (0–4)	**0.045** (0.053/0.34/1)
	I-Q	0.3 ± 0.5 (0–1.5)	6.5 ± 5.4 (0.5–17.0)	8.6 ± 7.7 (0–18.7)	**0.02** (**0.02**/0.30/1)
AANF deposition severity					
AS		1.4 ± 1.0 (0–3)	1.8 ± 1.0 (1–4)	1.3 ± 0.9 (0–2)	0.86 (1/1/1)
RA		1.5 ± 0.9 (0–3)	1.3 ± 0.7 (0–2)	1.3 ± 0.5 (1–2)	0.91 (1/1/1)
RAA		2.0 ± 1.0 (0–3)	1.7 ± 0.5 (1–2)	2.0 ± 0.8 (1–3)	0.53 (0.84/1/1)
LA		1.9 ± 1.1 (0–4)	1.7 ± 0.7 (1–3)	2.7 ± 0.5 (2–3)	0.16 (1/0.41/0.17)
LAA		2.3 ± 0.8 (1–3)	2.1 ± 1.0 (1–4)	2.3 ± 0.5 (2–3)	0.87 (1/1/1)
SVC		0.6 ± 0.7 (0–2)	0.7 ± 0.5 (0–1)	1.0 ± 0.0 (1)	0.52 (1/0.76/1)

**Boldface** signifies values that are statistically significant at *p* < 0.05. Abbreviations: N/PN, neural/perineural; Q, quantitative analysis result; SQ, semiquantitative analysis result. * Evaluated using the Kruskal–Wallis test with post-Bonferroni correction.

**Table 4 ijms-25-08176-t004:** Summary of the results of the semiquantitative and quantitative analyses for females and males.

Semiquantitative ATTR Deposition Grading (Range)				
Region		Female	Male	*p* Value *
Ventricular septum	Endocardium	0.1 ± 0.3 (0–1)	0.4 ± 0.7 (0–2)	0.37
	Vessel	0.9 ± 1.2 (0–3)	1.2 ± 1.4 (0–4)	0.66
	Interstitium	1.8 ± 1.1 (0–4)	3.0 ± 0.9 (1–4)	**0.04**
Atrial septum	Endocardium, right	2.0 ± 1.7 (0–4)	2.8 ± 1.3 (0–4)	0.37
	Endocardium, left	0.5 ± 0.9 (0–3)	0.7 ± 0.9 (0–3)	0.55
	Vessel	0.5 ± 0.7 (0–2)	0.9 ± 1.0 (0–3)	0.55
	Interstitium	1.7 ± 1.5 (0–4)	2.8 ± 1.2 (0–4)	0.15
Right atrium	Endocardium	0.7 ± 1.1 (0–3)	1.6 ± 1.3 (0–4)	0.13
	Vessel	0.9 ± 1.1 (0–3)	1.4 ± 1.1 (0–3)	0.30
	Interstitium	0.9 ± 1.1 (0–4)	1.3 ± 1.2 (0–4)	0.33
Right atrial appendage	Endocardium	0.7 ± 1.0 (0–3)	1.4 ± 1.2 (0–4)	0.18
	Vessel	0.9 ± 1.6 (0–4)	0.9 ± 0.9 (0–3)	0.41
	Interstitium	0.6 ± 0.6 (0–2)	1.0 ± 0.8 (0–4)	0.37
Left atrium	Endocardium	1.3 ± 1.0 (0–3)	2.1 ± 0.7 (1–3)	0.08
	Vessel	1.3 ± 1.2 (0–4)	1.7 ± 0.9 (1–3)	0.37
	Interstitium	2.0 ± 1.2 (1–4)	3.1 ± 1.1 (1–4)	0.07
Left atrial appendage	Endocardium	0.7 ± 0.9 (0–3)	1.7 ± 1.4 (0–4)	0.13
	Vessel	0.8 ± 1.2 (0–3)	1.1 ± 1.2 (0–4)	0.46
	Interstitium	1.9 ± 1.2 (1–4)	2.1 ± 0.6 (1–3)	0.37
Superior vena cava	Endocardium	1.5 ± 1.5 (0–4)	2.3 ± 1.4 (0–4)	0.30
	Vessel	0.5 ± 0.8 (0–2)	0.7 ± 1.1 (0–3)	0.77
	Interstitium	1.7 ± 1.2 (0–4)	1.6 ± 1.1 (0–4)	0.94
Semiquantitative AANF deposition grading (range)				
Region		Female	Male	*p* value
Atrial septum	Interstitium	1.5 ± 1.2 (0–4)	1.7 ± 0.8 (1–3)	0.71
Right atrium	Interstitium	1.5 ± 0.7 (0–2)	1.3 ± 0.8 (0–3)	0.30
Right atrial appendage	Interstitium	2.0 ± 0.9 (0–3)	1.7 ± 0.7 (1–3)	0.33
Left atrium	Interstitium	1.8 ± 0.7 (0–3)	2.0 ± 1.1 (1–4)	0.60
Left atrial appendage	Interstitium	2.4 ± 0.8 (1–3)	2.0 ± 0.9 (1–4)	1.00
Superior vena cava	Interstitium	0.6 ± 0.5 (0–1)	0.8 ± 0.6 (0–2)	0.71
Quantitative ATTR deposition burden (%)				
Region		Female	Male	*p* value
Ventricular septum	Interstitium	5.3 ± 6.9 (0–24.0)	12.8 ± 8.2 (0–24.9)	0.056
Atrial septum	Interstitium	5.5 ± 7.7 (0–20.8)	11.4 ± 7.5 (0–23.5)	0.10
Right atrium	Interstitium	1.6 ± 4.3 (0–15.2)	3.6 ± 6.1 (0–19.9)	0.20
Right atrial appendage	Interstitium	0.4 ± 1.1 (0–4.0)	2.1 ± 2.6 (0–7.1)	0.20
Left atrium	Interstitium	4.3 ± 4.7 (0.1–13.8)	10.6 ± 7.1 (0.1–18.4)	**0.04**
Left atrial appendage	Interstitium	5.4 ± 6.8 (0–18.5)	6.3 ± 4.3 (0.1–14.0)	0.37
Superior vena cava	Interstitium	4.5 ± 6.3 (0–18.7)	4.1 ± 5.0 (0–17.0)	0.50

**Boldface** signifies values are statistically significant at *p* < 0.05. * Female vs. male, compared using the Mann–Whitney U test.

## Data Availability

The datasets used and analyzed in the current study are available from the corresponding authors upon request.

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
