# Peer review of "Distinctive Deposition Patterns of Sporadic Transthyretin-Derived Amyloidosis in the Atria: A Forensic Autopsy-Based Study"

_ijms, 2024, doi:10.3390/ijms25158176_

Round 1

Reviewer 1 Report

Comments and Suggestions for Authors

The reviewer thanks the authors for their thorough submission and high technical skill. Though the techniques were executed with excellence, the clinical relevance of the article remains elusive to this particular reviewer. The below questions are listed to address this major deficit:

1. It seems that the authors' prior publication contain one of the main conclusion of the present article: 

"The atrial septum and basilar ventricular septum were the sites that revealed the most frequent deposition."

"In the atrial septum, although the severity of interstitial deposition was similar on both sides, high-grade endocardial deposition was significantly more common in the right side..."

Ichimata S, Hata Y, Hirono K, Yamaguchi Y, Nishida N. Clinicopathological features of clinically undiagnosed sporadic transthyretin cardiac amyloidosis: a forensic autopsy-based series. Amyloid. 2021 Jun;28(2):125-133. doi: 10.1080/13506129.2021.1882979. Epub 2021 Feb 8. PMID: 33554665.

Could the authors highlight the novelty of the present article compared with the prior? Though the present submission highlights left-to-right atrial deposition differences in wild type TTR, it is not clear that a new publication on a very similar topic is entirely warranted.

2. Could the authors elaborate on the limitations of the present article, given very few are provided outside of line 239 and many exist.

3. Can the authors elaborate on the clinical/real-world utility of the present submission? As in, how can this manuscript potentially useful in improving patient outcomes

Author Response

Response to Reviewer #1

We are grateful for your kind review and comments, which have helped us substantially improve our manuscript. Additions are indicated in red font in the revised manuscript.

The reviewer thanks the authors for their thorough submission and high technical skill. Though the techniques were executed with excellence, the clinical relevance of the article remains elusive to this particular reviewer. The below questions are listed to address this major deficit:

  1. It seems that the authors' prior publication contains one of the main conclusions of the present article: "The atrial septum and basilar ventricular septum were the sites that revealed the most frequent deposition." "In the atrial septum, although the severity of interstitial deposition was similar on both sides, high-grade endocardial deposition was significantly more common in the right side..."Ichimata S, Hata Y, Hirono K, Yamaguchi Y, Nishida N. Clinicopathological features of clinically undiagnosed sporadic transthyretin cardiac amyloidosis: a forensic autopsy-based series. Amyloid. 2021 Jun;28(2):125-133. doi: 10.1080/13506129.2021.1882979. Epub 2021 Feb 8. PMID: 33554665. Could the authors highlight the novelty of the present article compared with the prior? Though the present submission highlights left-to-right atrial deposition differences in wild type TTR, it is not clear that a new publication on a very similar topic is entirely warranted.
  2. Could the authors elaborate on the limitations of the present article, given very few are provided outside of line 239 and many exist.
  3. Can the authors elaborate on the clinical/real-world utility of the present submission? As in, how can this manuscript potentially useful in improving patient outcomes.

Response:

Thank you for these suggestions.

  1. We previously reported that the severity of deposition differed between the right and left atrial septal endocardia. Herein, we have demonstrated significant differences in the amount of deposition in the atrial myocardium, which is most involved in atrial function. Therefore, the results of the present study may be more directly related to atrial function than previous findings and are likely to be clinically important. We have included this description in the revised manuscript (lines 316–320).
  2. We have included a description of some limitations in this study (lines 306–314).
  3. We have added descriptions on the clinical/real-world utility to the revised manuscript (lines 239–242 and 330–333). Furthermore, since there are few publications showing histopathology of amyloid deposits in peripheral nerves in ATTR amyloidosis, we have added a new Figure 1, which is representative of the findings in this study (Page 4).

Reviewer 2 Report

Comments and Suggestions for Authors

The title clearly states the focus of the study on the left-to-right differences in transthyretin-derived amyloid (ATTR) deposition in the atria of older adults. Results are presented clearly, highlighting the significant findings of more severe ATTR deposition in the left atrial regions and among males.

  • The discussion could better address study limitations, such as the sample size and generalizability to broader populations.
  • Comparisons with other studies on ATTR deposition in different cardiac regions could provide a more comprehensive context.
  • Future research directions should be more explicitly stated to guide subsequent investigations.

Comments on the Quality of English Language

Minor editing of English language required. 

Author Response

Response to Reviewer #2

Thank you for your kind review and comments, which have been very helpful in improving our manuscript. In the revised manuscript, additions are indicated in red font.

The title clearly states the focus of the study on the left-to-right differences in transthyretin-derived amyloid (ATTR) deposition in the atria of older adults. Results are presented clearly, highlighting the significant findings of more severe ATTR deposition in the left atrial regions and among males.

  1. The discussion could better address study limitations, such as the sample size and generalizability to broader populations.
  2. Comparisons with other studies on ATTR deposition in different cardiac regions could provide a more comprehensive context.
  3. Future research directions should be more explicitly stated to guide subsequent investigations.

Response:

Thank you for your insights.

  1. We have described some limitations in this study (lines 306–314).
  2. As far as we have reviewed, we have not found reports on the histopathological findings of ATTR-CA deposition patterns in regions other than the ventricles. Therefore, we have cited and discussed a report that showed a difference in the dysfunction degree of the left atrium between AL-CA and ATTR-CA (lines 239–242; ref#25).
  3. We believe that the distinction between ATTR-CA and AL-CA is crucial, particularly for treatment. Currently, the histopathological examination of deposition patterns in the right and left atria in AL-CA cases remains unreported, and we believe that this issue should be addressed in the future. We have added a description regarding this matter to the revised manuscript (lines 330–333). Furthermore, since there are few publications showing histopathology of amyloid deposits in peripheral nerves in ATTR amyloidosis, we have added a new Figure 1, which is representative of the findings in this study (Page 4).

Round 2

Reviewer 1 Report

Comments and Suggestions for Authors

The reviewer thanks the authors for their thorough revisions and figure addition. Other concerns that exist upon repeat review:

1. For completeness sake, can it be reiterated what role Dr. Keiichi Hirono (affiliation #2, Pediatrics) fulfilled in the present article with regard to methodology and manuscript revisions? Given ATTR is a disease of senescence and likely lies outside the expertise of pediatric medicine

2. What is meant by "constipation" (line 196)? 

3. Please provide the definition of all abbreviations utilized in figure 3. This request applies broadly to all figures and tables in the manuscript, please carefully review as some legend abbreviations lists are incomplete.

4. Does the present study supersede/contradict the group's prior (Clinicopathological features of clinically undiagnosed sporadic transthyretin cardiac amyloidosis) findings regarding the distribution of misfolded protein within the atria? There is no mention of the septal/endocardial density in the present publication to this reviewer's understanding

5. Was the "Semiquantitative grading system for ATTR and AANF deposition" completed with in an automated/artificial intellegence-assisted manner? Or manually? If manual, please provide a comprehensive list of the details regarding the observer(s) and blinding, and add this to the list of limitations. Furthermore, it would be ideal to have all of the limitations within a single paragraph.

Author Response

Response to Reviewer #1

We are grateful for your kind review and comments, which have helped us substantially improve our manuscript. Additions are indicated in red font in the revised manuscript.

  1. For completeness sake, can it be reiterated what role Dr. Keiichi Hirono (affiliation #2, Pediatrics) fulfilled in the present article with regard to methodology and manuscript revisions? Given ATTR is a disease of senescence and likely lies outside the expertise of pediatric medicine.
  2. What is meant by "constipation" (line 196)? 
  3.  Please provide the definition of all abbreviations utilized in figure 3. This request applies broadly to all figures and tables in the manuscript, please carefully review as some legend abbreviations lists are incomplete.
  4. Does the present study supersede/contradict the group's prior (Clinicopathological features of clinically undiagnosed sporadic transthyretin cardiac amyloidosis) findings regarding the distribution of misfolded protein within the atria? There is no mention of the septal/endocardial density in the present publication to this reviewer's understanding.
  5. Was the "Semiquantitative grading system for ATTR and AANF deposition" completed with in an automated/artificial intellegence-assisted manner? Or manually? If manual, please provide a comprehensive list of the details regarding the observer(s) and blinding, and add this to the list of limitations. Furthermore, it would be ideal to have all of the limitations within a single paragraph.

  1. Dr. Keiichi Hirono, a cardiologist, interpreted how the results of this experiment could be useful for clinical testing and reviewed the paper based on his interpretation.
  2. We have corrected the typo (Lines 195–196).
  3. We have added the definition of all abbreviations (Lines 141–142, and 493–498).
  4. Our previous study was based on a proximity ligation assay. In the present study, we tried to prove it with IHC and were unable to do so. Therefore, as discussed in Lines 258–260, we believe that the only way to reach a conclusion is to conduct in vitro experiments. With regard to the endocardium, the present study also showed a right-predominant deposition pattern similar to our previous results. We have added this information in Lines 154–156.
  5. Selection of ATTR deposition-positive cases was performed by one cardiovascular pathologist (N.N) based on the histopathological and IHC findings, and the semiquantitative ATTR and AANF deposition burden analysis was performed separately by another cardiovascular pathologist (S.I.). Thus, in the latter processes, all assessments were performed blinded to clinical information and pathological diagnoses. We have added this information in Lines 316–320.

Round 3

Reviewer 1 Report

Comments and Suggestions for Authors

The reviewer thanks the authors for their revisions and brief explanations. Additional comments are as follows:

1. The statistical analysis methods are distributed throughout the manuscript but are not appropriately consolidated within the methods section. Furthermore, essential details are missing, such as whether tests were run in a single-tail or two-tailed approach, given it seems that the authors' hypothesis was that a difference would exist between the two atria. This also illustrates the manuscript's need for an explicit hypothesis.

Author Response

Response to Reviewer #1

We thank the Reviewer for their kind review and comments, which have helped us substantially improve our manuscript. Additions are indicated in red font in the revised manuscript.

1. The statistical analysis methods are distributed throughout the manuscript but are not appropriately consolidated within the methods section. Furthermore, essential details are missing, such as whether tests were run in a single-tail or two-tailed approach, given it seems that the authors' hypothesis was that a difference would exist between the two atria. This also illustrates the manuscript's need for an explicit hypothesis.

Response:

We thank the Reviewer for these suggestions.

1. We have added descriptions of the analytical methods used for statistical analysis (lines 143, 202, 473–479).